# Nickel-catalyzed switchable arylative/*endo*-cyclization of 1,6-enynes

Wenfeng Liu[1,3], Wei Li[1,3], Weipeng Xu[2], Minyan Wang [2] ✉ & Wangqing Kong [1] ✉

Carbo- and heterocycles are frequently used as crucial scaffolds in natural products, fine chemicals, and biologically and pharmaceutically active compounds. Transition-metal-catalyzed cyclization of 1,6-enynes has emerged as a powerful strategy for constructing functionalized carbo- and heterocycles. Despite significant progress, the regioselectivity of alkyne functionalization is entirely substrate-dependent. And only *exo*-cyclization/cross-coupling products can be obtained, while *endo*-selective cyclization/cross-coupling remains elusive and still poses a formidable challenge. In this study, we disclose a nickel-catalyzed switchable arylation/cyclization of 1,6-enynes in which the nature of the ligand dictates the regioselectivity of alkyne arylation, while the electrophilic trapping reagents determine the selectivity of the cyclization mode. Specifically, using a commercially available 1,10-phenanthroline as a ligand facilitates *trans*-arylation/cyclization to obtain seven-membered ring products, while a 2-naphthyl-substituted bisbox ligand promotes *cis*-arylation/cyclization to access six-membered ring products. Diastereoselective cyclizations have also been developed for the synthesis of enantioenriched piperidines and azepanes, which are core structural elements of pharmaceuticals and natural products possessing important biological activities. Furthermore, experimental and density functional theory studies reveal that the regioselectivity of the alkyne arylation process is entirely controlled by the steric hindrance of the ligand; the reaction mechanism involves *exo*-cyclization followed by Dowd-Beckwith-type ring expansion to form *endo*-cyclization products.

The structural diversity of core molecular scaffolds is considered a critical feature of compound libraries, which is decisive for ensuring their success in drug discovery and biological research[1–3]. Compound libraries constructed by traditional synthetic strategies are mainly composed of molecules with similar structures. This is because the structural diversity of the products depends only on the starting scaffolds and building blocks. Therefore, the development of new strategies to efficiently construct structurally distinct and diverse molecular scaffolds, thereby giving rise to compound libraries with differing and selective biological activities is of great siginificance[4,5]. Ligand-directed divergent synthesis (LDS) is highly sought-after in library synthesis because it allows the construction of structurally diverse molecular scaffolds from common starting materials by simply changing the ligands of metal catalysts[6]. However, it is undoubtedly the most challenging strategy because ligand design alone requires subversion of the inherent regiopreferences of reactants and metal centers.

[1]The Institute for Advanced Studies (IAS), Wuhan University, Wuhan 430072, China. [2]State Key Laboratory of Coordination Chemistry, School of Chemistry and Chemical Engineering, Nanjing University, Nanjing 210023, China. [3]These authors contributed equally: Wenfeng Liu, Wei Li. ✉e-mail: wangmy@nju.edu.cn; wqkong@whu.edu.cn

Cyclic scaffolds, especially heterocycles, are ubiquitous in natural products, pharmaceuticals, and numerous functional molecules. The development of effective methods for the rapid construction of various cyclic compounds represents one of the major domains of synthetic organic chemistry. In this regard, transition-metal-catalyzed cycloisomerization[7–11] and reductive cyclization[12–16] of 1,6-enynes have been established to generate five-membered (hetero)cycles. Advances along this line have revealed some intriguing transition metal-catalyzed functionalizing cyclization reactions, providing efficient and convenient access to a wide array of elusive cyclic systems[17–34]. The appeal of these cyclizations lies in the streamlining construction of structurally complex and functionalized cyclic scaffolds from readily accessible 1,6-enynes. From a reaction mechanism perspective, these transformations are primarily initiated by the formation of metalacyclic intermediates via oxidative cyclization pathways or the formation of alkenyl metal species via carbometallation processes. The former has been well explored[7–16], but controlling the regioselectivity of the latter is difficult, usually in two ways: α-insertion or β-insertion (Fig. 1a)[35–38]. 5-*Exo*-cyclization of the resulting alkenyl-metal intermediate **I** (via β-insertion) would provide the conventional five-membered ring product **III** bearing an exocyclic double bond. Alternatively, α-insertion followed by *E/Z* isomerization[39–44] would afford alkenyl-metal intermediate **II**, which undergoes 6-*exo* cyclization to afford six-membered ring product **V** possessing an endocyclic double bond. Despite enormous efforts, these strategies still suffer from many limitations: the regioselectivity of alkynes insertion into organometallic species is entirely substrate-dependent (electronic or steric effects of substituents on alkynes). Furthermore, since *exo*-cyclizations are anyway kinetically superior to the corresponding *endo*-cyclizations, 6-*endo*- or 7-*endo*-cyclization products are rarely obtained[45–48].

Nickel-catalyzed reductive cross-coupling has been emerged as a powerful and straightforward method for constructing carbon-carbon bonds[49–54]. Furthermore, due to its unique properties, nickel has been used in a range of ligand-controlled divergent transformations[55–57]. To address the major remaining challenges of 1,6-enyne cyclization, we were curious whether changing the structure and/or electronegativity of the ligands in nickel catalysts would enable regioselective and switchable arylation of alkynes. Here, we demonstrate that judicious choice of ligands enables the switchable arylation/cyclization of 1,6-enynes, in which the nature of the ligand dictates the regioselectivity of alkyne arylation, while the electrophilic trapping reagents determine the selectivity of the cyclization mode. Specifically, in the presence of electron-deficient alkenes, using a commercially available 1,10-phenanthroline as a ligand facilitates *trans*-arylation/*endo*-cyclization to obtain seven-membered ring products, while a 2-naphthyl-substituted Bisbox ligand promotes *cis*-arylation/*endo*-cyclization to access six-membered ring products (Fig. 1b). Diastereoselective cyclizations have also been developed for the synthesis of enantioenriched piperidines and azepanes, which are core structural elements of pharmaceuticals and natural products possessing important biological activities. This transformation is modular, with the ligand rather than the substrate precisely determining the regioselectivity of alkyne arylation process, and the electrophilic trapping reagents determining the regioselectivity of cyclization. Such ligand-directed divergent synthesis (LDS) remains hitherto elusive despite persistent interest.

## Results and discussion
### Reaction development
To test our hypothesis, we started our investigation by evaluating the nickel-catalyzed three-component cascade cyclization of 1,6-enyne (**1**) with PhBr (**2**) and trifluoromethyl alkene (**3**) (Table 1). After a series of experiments, a combination of NiBr$_2$(DME), 1,10-phenanthroline **L4**, Mn, CsF, and DMA at 60 °C afforded *gem*-difluoroallylated seven-membered ring product **4** in 63% isolated yield with exquisite regioselectivity (**4**/**5** = 20/1) (entry 1). Inferior results were observed when either TBAI or NaI was used instead of CsF (entries 2 and 3). Lowering the reaction temperature to 40 °C resulted in no reaction, while increasing the reaction temperature to 80 °C resulted in lower yield (entries 4 and 5). Traces, if any, of **4** were observed when Zn was used instead of Mn as the terminal reducing agent (entry 6). Notably, the direct reductive coupling of aryl bromides and trifluoromethyl alkenes in this catalytic system is well suppressed.

We evaluated a range of different ligands with the goal of identifying the conditions for switchable access to the seven-membered ring product **4** and the six-membered ring product **5**. 2,2′-Bipyridine (**L1-L3**) and 1,10-phenanthroline (**L4**) favored the formation of **4**, especially the latter providing **4** in higher yield and excellent regioselectivity (entries 7–10). Strikingly, under the same reaction conditions as shown in entry 10, the use of relatively electron-rich Pyrox **L5** or Bisbox **L6** as ligands resulted in a regioselective switch en route to **5**, albeit in lower yields (entries 11 and 12). We further synthesized a series of Bisbox ligands with different substituents on the oxazoline ring and examined their performances. The phenyl-substituted ligand **L7** showed slightly higher regioselectivity (**4**/**5** = 1/8, entry 13). The benzyl-substituted ligand **L8** provided **5** in 58% yield and improved the regioselectivity to 1:15 (entry 14). Using a 2-naphthyl-substituted Bisbox ligand **L9**, **5** was obtained in 71% isolated yield and excellent regioselectivity (**4**/**5** = 1:28) (entry 15). These results clearly demonstrate that the backbone of the ligand determines the regioselectivity pattern.

Having identified optimal reaction conditions for the ligand-controlled divergent synthesis of seven- and six-membered ring

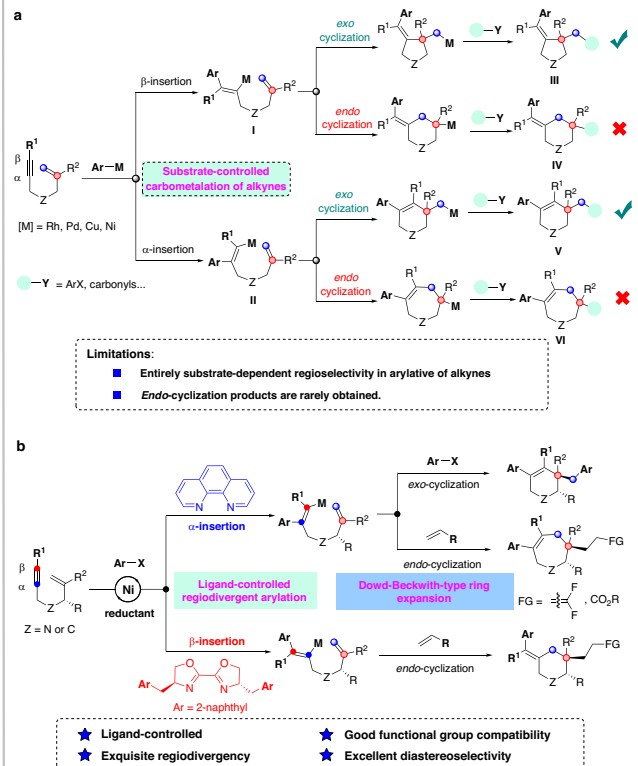

**Fig. 1 | Transition metal-catalyzed functionalization/cyclization of 1,6-enynes. a** The state of the art of transition metal-catalyzed arylative cyclization of 1,6-enynes; **b** Ligand-controlled regiodivergent arylative cyclization of 1,6-enynes.

## Table 1 | Optimization of reaction conditions

| entry | deviation from standard conditions | yield (%)[a] | 4/5[b] |
|---|---|---|---|
| 1 | CsF | 70 (63) | 20/1 |
| 2 | TBAI instead of CsF | 48 | 15/1 |
| 3 | NaI instead of CsF | 54 | 15/1 |
| 4 | 40 °C | n.r. | – |
| 5 | 80 °C | 30 | 15/1 |
| 6 | Zn instead of Mn | trace | – |
| 7 | L1 instead of L4 | 21 | 2/1 |
| 8 | L2 instead of L4 | 23 | 7/1 |
| 9 | L3 instead of L4 | 31 | 11/1 |
| 10 | none | 46 | 15/1 |
| 11 | L5 instead of L4 | 17 | 1/1.2 |
| 12 | L6 instead of L4 | 33 | 1/6 |
| 13 | L7 instead of L4 | 41 | 1/8 |
| 14 | L8 instead of L4 | 58 | 1/15 |
| 15 | L9 instead of L4 | 75 (71) | 1/28 |

Reaction conditions: **1** (0.2 mmol), **2** (0.2 mmol), **3** (0.1 mmol), NiBr$_2$(DME) (0.01 mmol), ligand (0.02 mmol), Mn$^0$ powder (0.3 mmol), and additive (0.1 mmol) in DMA (2 mL) in sealed tube at 60 °C for 48 h.
[a]Yields were determined by $^{19}$F NMR spectroscopy using trifluorotoluene as the internal standard. The values in brackets corresponds to the yields of isolated products after column chromatography on silica gel.
[b]Ratio determined by $^{19}$F NMR spectroscopy.

products, we turned our attention to investigating the scope and limitations of those two protocols. We first explored the *trans*-arylative/*endo* cyclization protocol for the selective synthesis of seven-membered ring products using ligand **L4** (Fig. 2). In terms of 1,6-enynes, the carbon-tethered and nitrogen-tethered 1,6-enynes were examined, and the corresponding products **6** and **7** were obtained as expected. The molecular structure of **7** was unambiguously confirmed by single-crystal X-ray diffraction analysis. Electron-donating (OMe) or electron-withdrawing groups (CF$_3$, CN) on the aromatic ring at the alkyne terminal (R$^1$) were found to be compatible, leading to the corresponding products **8-10** in 49-69% yields. Notably, chlorine substitution on the aromatic ring also proceeded smoothly, affording **11** in 68% yield, providing an opportunity for further derivatization of the obtained product. Complex molecules such as

estrone could also be successfully incorporated into product (**12**). Nonetheless, alkyl-substituted alkynes and terminal alkynes failed to produce the expected products. Introduction of a phenyl group on the alkene moiety did not preclude the transformation (**13**).

We then evaluated the scope of aryl (pseudo)halides. We were pleased to find that the presence of electron-donating and electron-withdrawing groups at the *para* or *meta* position of the aromatic ring did not affect selectivity and efficiency (**14–23**). Aryl triflate was also tolerated as shown by the efficient formation of product **15**. Heteroaryl bromides such as 5-bromo-2-methoxypyridine, 3-bromothiophene, 2-bromodibenzofuran, 2-bromodibenzothiophene, 5-bromobenzo-furan, 5-bromoindole, and 3-bromocarbazole were all efficiently transformed into the corresponding products (**24–30**) in satisfactory yields.

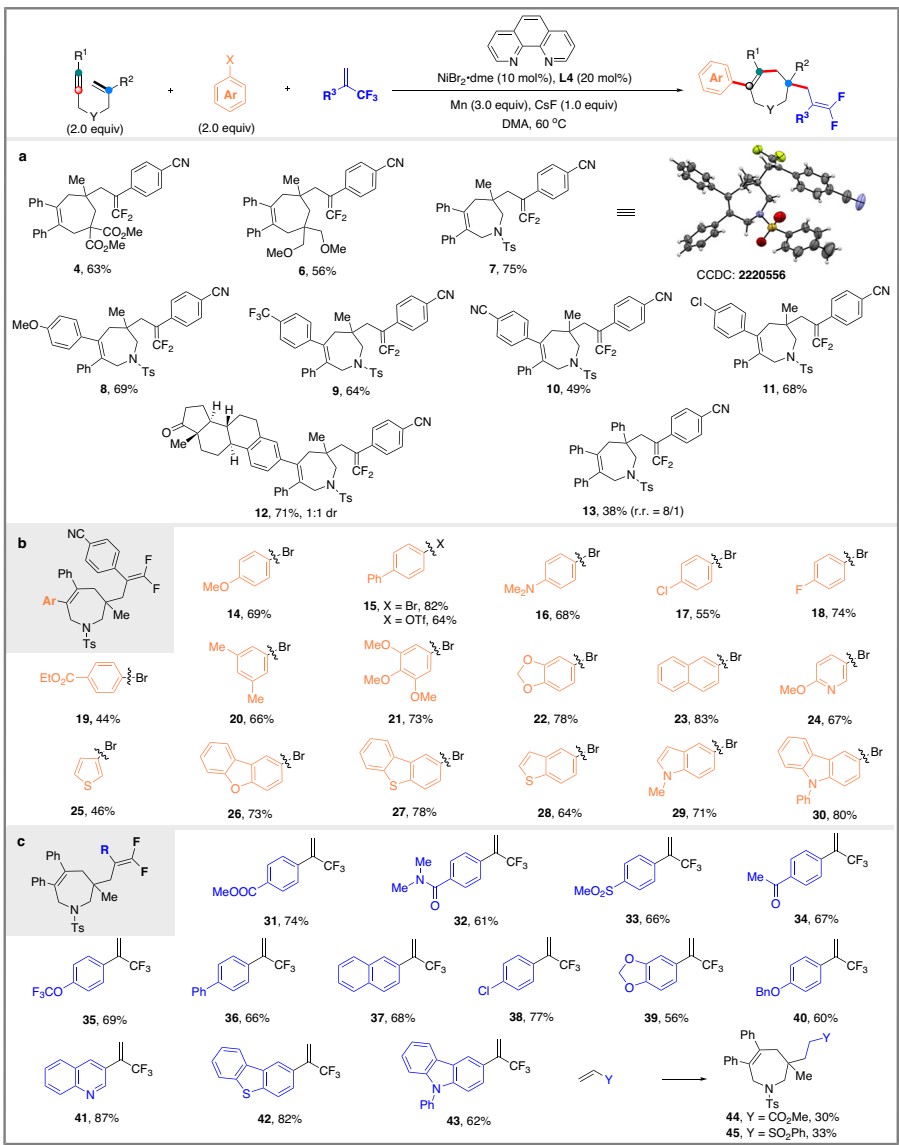

**Fig. 2 | Scope for *trans*-arylative cyclization. a** Scope of 1,6-enynes; **b** Scope of aryl halides; **c** Scope of electron-deficient alkenes.

The generality of the electron-deficient alkenes was also investigated. Trifluoromethyl alkenes with various useful functional groups, such as cyano, ester, amide, sulfonate, trifluoromethoxy, chloro, and alkoxy on the aromatic ring, could be successfully transferred to the corresponding products (**31**–**40**). In addition, quinoline, benzothiophene and carbazole-substituted trifluoromethyl alkenes (**41**, **42** and **43**) were also compatible with the reaction conditions. The seven-membered ring product **44** and **45** were also obtained under the same reaction conditions when electron-deficient alkenes such as acrylate or vinyl sulfone were used respectively.

We next examined the scope of the *cis*-arylative/*endo* cyclization protocol for the selective synthesis of six-membered ring products using ligand **L9** (Fig. 3). Similar to what we observed with the *trans*-arylative/*endo* cyclization, C- or N-tethered 1,6-enynes had little effect on the reactivity and selectivity, providing the corresponding products (**46**-**47**) in 58–63% yields. 1,6-Enynes with substituents on the alkene moiety such as phenyl and ester were also compatible (**48**-**49**). Both electron-withdrawing (Cl) and electron-donating groups (OMe) on the aromatic ring at the alkyne terminal (R¹) were found to be compatible (**50**-**51**).

Aryl (pseudo)halides with versatile functional groups were then examined. It is noteworthy to mention that substrates with both electron-donating and withdrawing groups proceed smoothly to afford the corresponding products (**52**-**58**) in moderate to good yields. In particular, the *p*-chlorine substituted product **54** and the 3-bromocarbazole substituted product **59** were both isolated in good yields.

In addition, the compatibility of the transformation with various trifluoromethyl alkenes was evaluated in a robustness screening. A series of valuable functional groups such as cyano, ester, amide, sulfonate, ketone, trifluoromethoxy, chloro, and alkoxy were all well-tolerated (**60**–**69**). Trifluoromethyl alkenes bearing heterocyclic groups could also be transformed into the desired products **70**–**73** in moderate yields. In the case of the trifluoromethylalkenes with a strong electron-donating group, the desired defluorinative products **68**–**72** were isolated in a mixture with the reductive Heck coupling products which retained the trifluoromethyl group.

Enantiomerically pure *N*-heterocyclic compounds, especially piperidines and azepanes, represent important motifs in pharmaceutically important natural alkaloids[58]. Consequently, it is of great significant to develop reliable synthetic methods for the synthesis of such

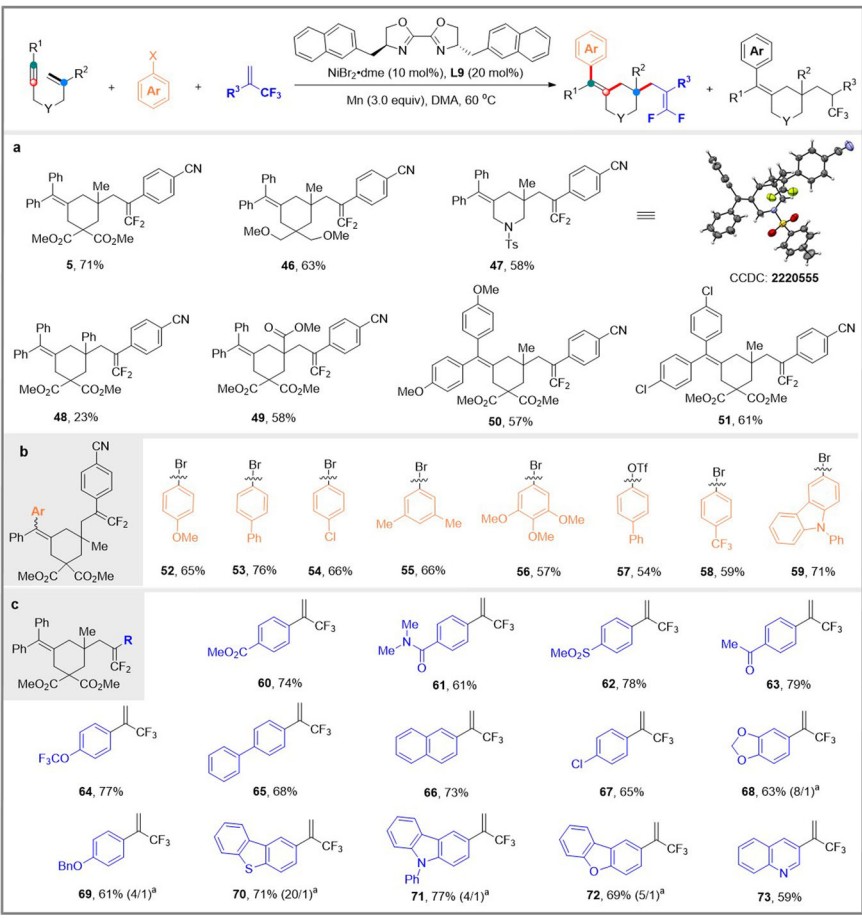

**Fig. 3 | Scope for *cis*-arylative cyclization. a** Scope of 1,6-enynes; **b** Scope of aryl halides; **c** Scope of electron-deficient alkenes. [a]The ratio of product to reduced Heck product was determined by [19]F NMR spectroscopy.

heterocycles. We screened a series of chiral ligands in an attempt to render the reaction asymmetric (Supplementary Table 1). Simultaneously controlling the regioselectivity of alkyne migratory insertion and the stereoselectivity of the cyclization process is indeed a very challenging task, and all our attempts resulted in only racemic products.

We turned to developing a solution to obtain enantioenriched piperidines and azepanes by diastereoselective cyclization of substrates with pre-existing stereocenters[59,60]. Enantioenriched 1,6-enyne **74** (e.r. > 99:1) was prepared employing Ellman's auxiliary[61] to test subsequent ligand-controlled arylative cyclization for the divergent synthesis of chiral piperidines and azepanes (Fig. 4a). Excitingly, by a simple ligand switch, enantioenriched piperidines **75–82** and azepanes **83–86** can be obtained, respectively, from the same starting materials in synthetically useful yields and excellent diastereoselectivities (15:1 to > 20:1 d.r.). The absolute configuration of **76** was determined by single-crystal X-ray diffraction analysis, and that of all other products were assigned accordingly.

We further found that the reaction of 1,6-enyne with aryl bromide underwent *trans*-arylative *exo*-cyclization/cross-coupling to give diarylated products when 1,10-phenanthroline **L4** was used as a ligand (Fig. 4b). The presence of electron-donating and electron-withdrawing groups at the *para*- or *meta*-position of the aryl bromides does not affect selectivity and efficiency (**87–95**). Diastereoselective diarylative cyclization gave enantioenriched piperidine **96** in excellent diastereoselectivity (> 20:1 d.r.). Despite extensive attempts, *cis*-arylation cyclization/cross-coupling products have not yet been obtained using **L9** as a ligand.

## Mechanistic investigation

To shed light on the reaction mechanism, we designed a series of experiments as outlined in Fig. 5. First, Ph-Ni(II)Br complex **98** and **99** were synthesized and submitted to the reactions with 1,6-enyne **97** and trifluoromethyl alkene **3** under our standard reaction conditions. The corresponding products **7** and **47** were obtained in 31% and 30% yields, respectively (Fig. 5a). These results clearly demonstrate that active aryl-nickel(II) species are generated during the reaction and that the ligand determines the regioselectivity alkyne arylation.

We further synthesized alkenyl iodide **100** and subjected to the reaction with trifluoromethylalkene **3** under standard conditions. The target product **7** was isolated in 65% yield, evidently confirming the formation of the key alkenyl-nickel intermediate and the involvement of *cis/trans* isomerization of alkenyl-nickel species during the reaction (Fig. 5b). Deuterium labeling experiments were performed. Reaction of **100** with 5 equivalents of D$_2$O afforded the deuterated *exo*-cyclization product **101-D** in 43% yield. Reaction of **100** with trifluoromethylalkene **3** in the presence of 10 equivalents of D$_2$O gave product **101-D** and **7** in 24% and 49% yield, respectively (Fig. 5c). These deuterated products should result from deuteration of the corresponding organonickel intermediates, suggesting that both 6-*exo*-cyclization and 7-*endo*-cyclization of alkenyl-nickel intermediate appear to be possible.

Three possible reaction pathways are proposed to explain the transformation of six-membered-ring nickel intermediate **103** to seven-membered-ring nickel intermediate **106** in Fig. 5d. Considering that β-carbon elimination from a non-strained metal complex is a high

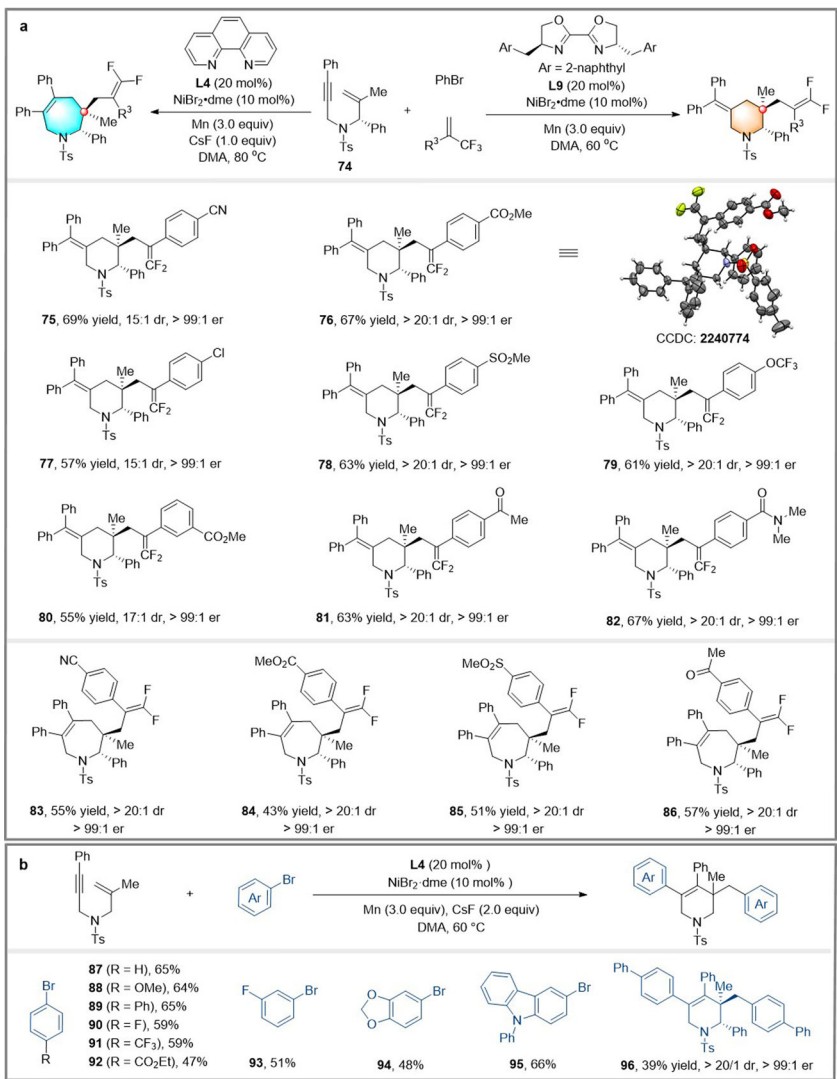

**Fig. 4 | Substrate scope. a** Diastereoselective synthesis of enantioenriched piperidines and azepanes; **b** Diarylative cyclization of 1,6-enynes.

energy process[62-64], a retro 6-*exo* cyclization via β-carbon elimination followed by 7-*endo* cyclization deemed less probable (Path I, Fig. 5d).

To verify 1,2-dyotropic rearrangement mechanism (Path II, Fig. 5d), deuterated 1,6-enyne (*Z*)-**107-D** was prepared and subjected to the reaction with trifluoromethylalkene **3** and PhBr under standard conditions. Product **7-D** was isolated in 55% yield with 1:1 diastereoselectivity (Fig. 5e). Although this process is non-stereoselective, considering the carbon-nickel bond is easily undergo homolytic cleavage to generate the corresponding carbon radical and then recombine, we cannot rule out the mechanism of 1,2-dyotropic rearrangement[65-67].

Furthermore, Dowd-Beckwith ring expansion process was also considered (Path III, Fig. 5d)[68,69], that is, the primary radical intermediate **104** and the tertiary radical intermediate **105** coexist in the reaction system; the type of electrophilic trapping reagents determine the structure of products. When cross-coupling with aryl bromides via an oxidative addition/reductive elimination mechanism, primary radicals **104** are preferred because the process is more sensitive to steric hindrance of ligands. The nucleophilic tertiary radicals **105** are favored when cross-coupling with less hindered electron-deficient alkenes (acrylates or trifluoromethyl alkenes) via Giese addition.

## DFT calculations

To better understand the reaction mechanism and the origin of selectivity-controlled steps of arylative of alkynes with different

ligands, density functional theory (DFT) studies were performed (Fig. 6)[70]. The computed energy profile for the nickel catalytic cycle using ligand **L4** is shown in Scheme 8 A, while the influence of ligand **L9** is shown in Supplementary Fig. 6. The overall pathway commences with the coordination of bromobenzene **2** to the in-situ generated Ni(0)/ligand species, forming intermediate **INT1A**. The oxidative addition of the C−Br bond proceeds through transition **TS2A** and subsequent substrate **1** replaces the Br⁻ ligand at **INT2A**, resulting in the formation of **INT3A**. The noncovalent interactions (NCI) between substrates and ligands is more pronounced for the regioselectivity of migratory insertion (Fig. 6b)[71,72]. In the case of smaller ligand **L4**, phenyl connected with nickel center and phenyl substituent of alkyne are the most sterically congesting moiety, thus resulting the energy barrier of transition state **TS4A-L4** is 4.9 kcal mol⁻¹ higher than that of transition state **TS4B-L4** (17.8 vs 12.9 kcal mol⁻¹). The α-migratory insertion to form intermediate **INT4B-L4** is kinetically more favorable pathway. For ligand **L9**, there is a significant attractive interaction between naphthyl group and phenyl unit of alkyne, rendering an increased energy barrier of **TS4B-L9** to 21.8 kcal mol⁻¹. This result illustrates intermediate **INT3A-L9** favors β-insertion over α-insertion. The resulting intermediate **INT4B-L4** undergoes rapid *cis/trans* isomerization via reversible Ni−C bond homolysis[73-75] or a η²-coordinated alkenylnickel species (see Supplementary Information for details)[76-78] to

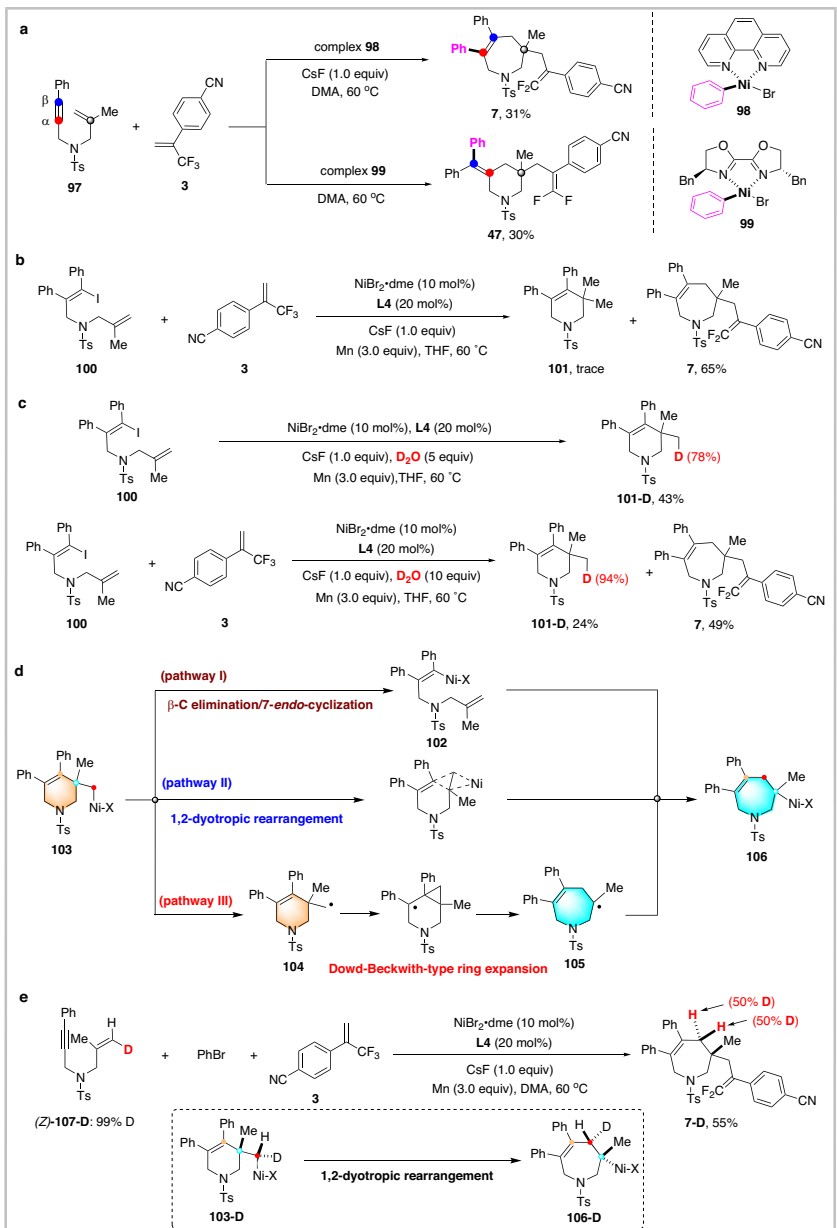

**Fig. 5 | Mechanistic studies. a** Reactions with arylnickel complex; **b** Reaction of trifluoromethyl alkene with alkenyl iodide; **c** Deuterium experiments; **d** Proposed reaction pathways; **e** Reaction of deuterated 1,6-enyne, aryl bromide and trifluoromethyl alkene.

form intermediate **INT5B-L4**, which facilitates the proximity of the nickel(II) center to the C = C bond for the subsequent cyclization process[35,79,80]. The competitive cyclization modes with ligand **L4** and **L9** are further investigated, respectively (Fig. 6c). Compared to the *endo*-cyclization, the *exo*-cyclization through **TS5B-L4-exo** and **TS5A-L9-exo** is a more favorable process (14.3 vs 11.3 kcal mol⁻¹ for **L4** and 19.4 vs 2.8 kcal mol⁻¹ for **L9**). Upon adding D₂O, the generated **INT5B-L4-exo** can be converted into six-membered skeleton smoothly, consistent with the experimental observation shown in Fig. 6c. Finally, the Ni(II)-alkyl bond of intermediate **INT5B-L4-exo** and **INT5A-L9-exo** undergoes homolytic cleavage to generate alkyl radical and Ni(I) species, which was then reduced by the present of Mn to regenerate **INT1A**, completing nickel catalytic cycle.

Furthermore, extensive computational studies on alkyl-nickel(II) species **INT5B-L4-exo** and **INT5A-L9-exo** were performed (Fig. 7a, b). We located a homolytic cleavage pathway of alkyl-nickel(II) to generate radical **INT7B** (Fig. 7a). By comparison, the alternative pathways

involving directly insertion into the double bond of trifluoromethyl alkene **3** (**TS6A-L4-exo** and **TS6A-L9-exo**, $\Delta G^{\ddagger} = 24.0$ and 33.0 kcal mol⁻¹, respectively), 1,2-alkenyl/Ni concerted dyotropic rearrangement (**TS6C-L4-exo** and **TS6C-L9-exo**, $\Delta G^{\ddagger} = 41.5$ and 47.2 kcal mol⁻¹, respectively), and 1,2-alkyl/Ni concerted dyotropic rearrangement (**TS6D-L4-exo** and **TS6D-L9-exo**, $\Delta G^{\ddagger} = 52.1$ and 57.7 kcal mol⁻¹, respectively), were calculated to be kinetically unfavorable.

The subsequent pathway for the generated radical **INT7B** was shown in Fig. 8. The radical induced Dowd-Beckwith-type ring expansion[68,81–85] through transition **TS8B** and **TS9B**, leading to the formation of seven membered ring radical **INT9B**. After that, **INT9B** captured by substrate **3** leads to the formation of **INT10B** with an activation free energy barrier of 7.9 kcal mol⁻¹. For the enantiomerically pure *N*-heterocyclic compound **72**, the radical addition preferentially occurred at the less steric hindrance to result excellent diastereoselectivities. Owing to the computed high energy barrier

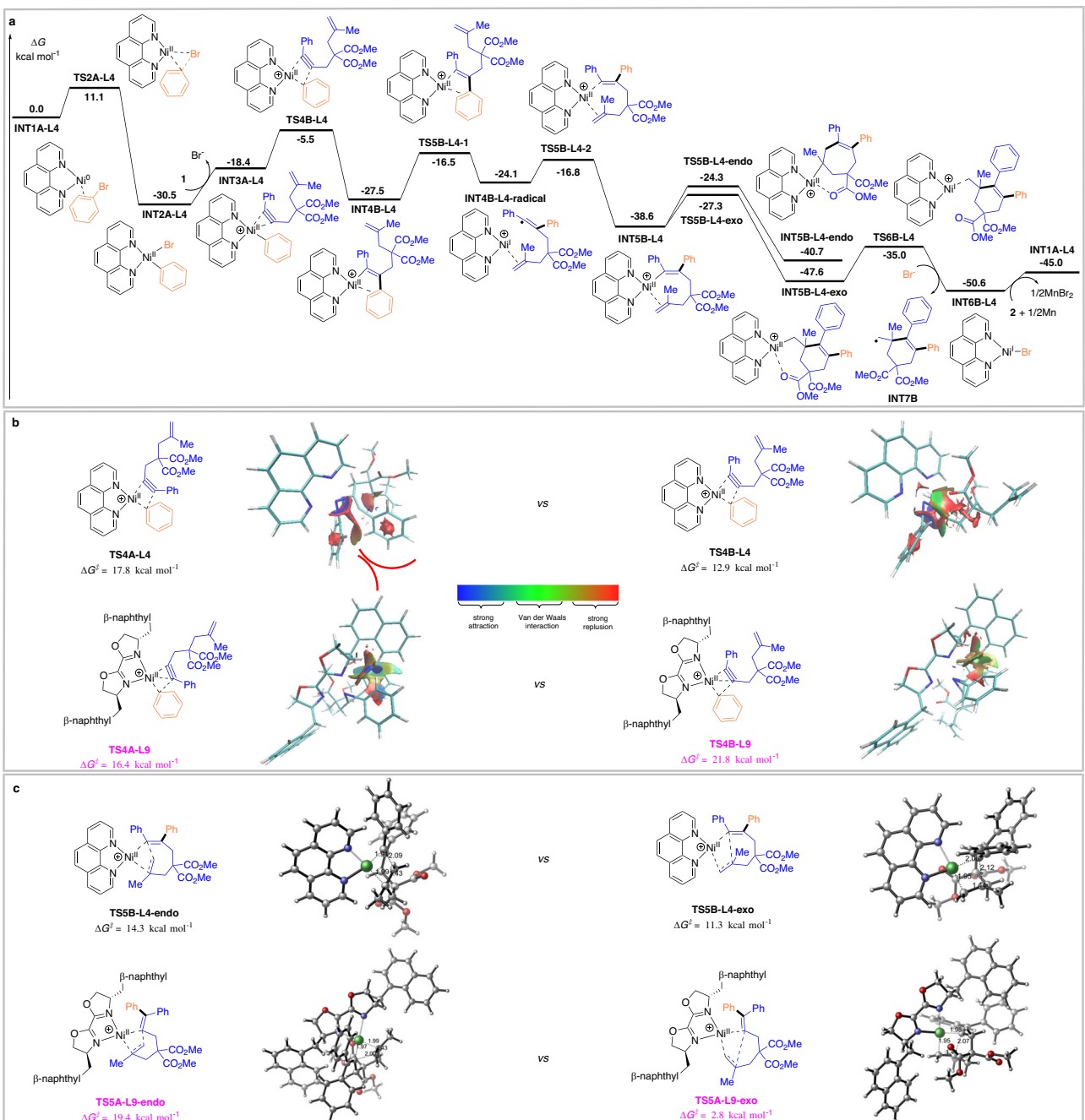

**Fig. 6 | DFT investigations. a** The calculated energy profile of the nickel-catalyzed cycle using ligand **L4**; **b** The competitive migratory insertion of aryl-nickel into alkyne with ligand **L4** and **L9**; **c** The competitive cyclization modes with ligand **L4** and **L9**.

(**TS11B**, $\Delta G^{\ddagger} = 29.6$ kcal mol⁻¹), the direct β-F elimination of **INT10B** is excluded. The single electron transfer (SET) process between **INT10B** and reductant Mn proceeds to afford anion **INT11B**, which is followed by β-F elimination via transition state **TS12B** with an activation energy of 9.9 kcal mol⁻¹, yielding the desired product **4**. The energy profile for the radical produced by **INT5A-exo-L9** is also calculated to obtain product **5** (for the details of the computational data, see Supplementary Fig. 8). The frontier molecular orbital theory is further used to analysis the reactivity of the generated radicals (Fig. 8c). Compared with radical **INT7B**, the radical **INT9B** with higher SOMO energy exhibits more nucleophilicity, which should be thermodynamically more favorable to attack electron-deficient alkenes to generate radical **INT10B**, delivering the desired cyclization products.

**Proposed reaction mechanism**

Based on the above experimental and DFT studies, a plausible catalytic cycle is outlined in Fig. 9. Oxidative addition of aryl bromide to Ni(0) species produces aryl-nickel(II) species. The steric interaction between the ligands and substrates precisely controls the regioselectivity of the migratory insertion of aryl-nickel(II) species into alkynes, where **L9** promotes β-insertion to form alkenyl-nickel(II) intermediate **A** and **L4** facilitates α-insertion to form alkenyl-nickel(II) intermediate **E**. Intermediates **A** and **E** undergo highly selective *exo*-cyclization to give alkyl-nickel(II) intermediates **B** and **F**, which then undergo homolytic cleavage to afford the corresponding primary alkyl radical intermediates **C** and **G**, respectively. Direct cross-coupling of radical **G** with aryl bromides in the presence of Ni(0)

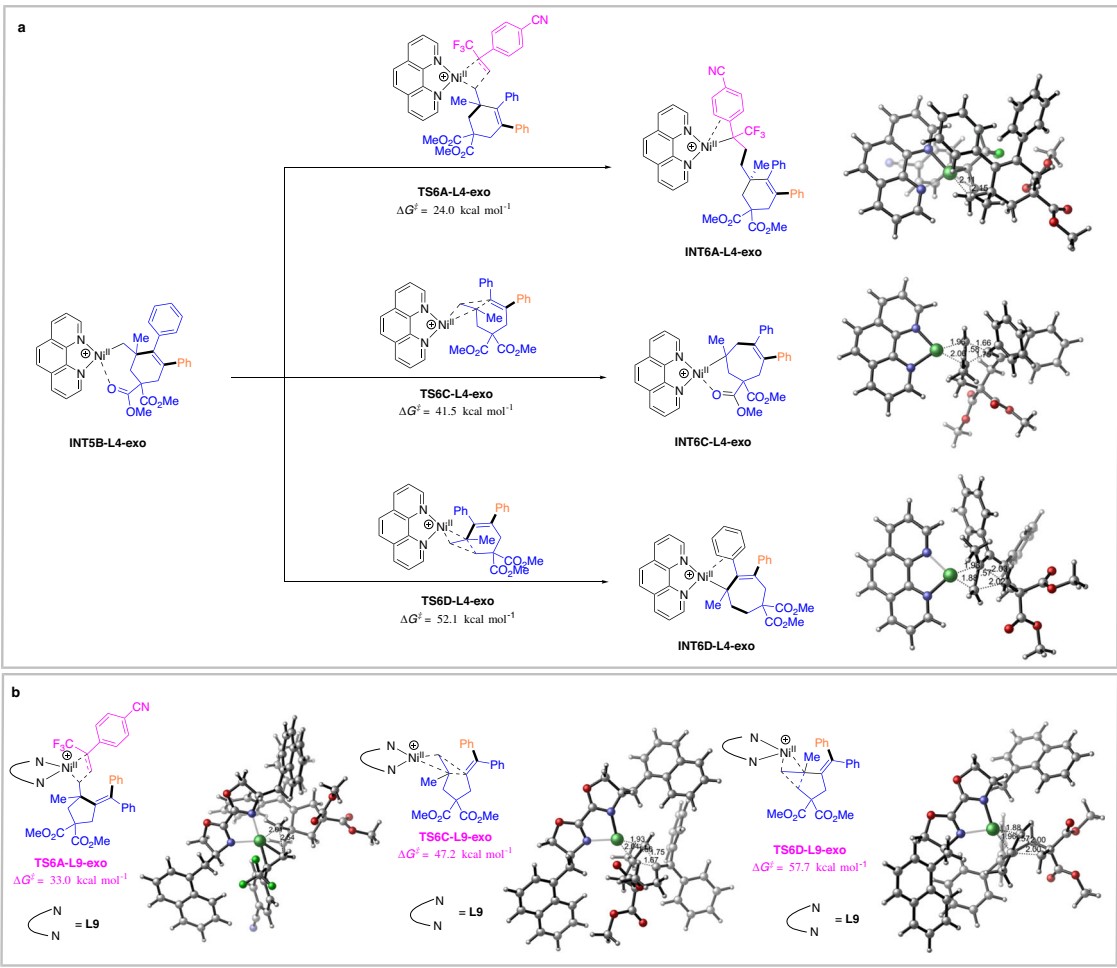

**Fig. 7 | Extensive computational studies. a** Extensive computational studies on alkyl-nickel(II) species **INT5B-L4-exo**; **b** Extensive computational studies on alkyl-nickel(II) species **INT5A-L9-exo**.

gives diarylated products. Intermediates **C** and **G** undergo Dowd-Beckwith ring expansion to form more stable tertiary radical intermediates **D** and **H**, respectively. Finally, radical addition followed by fluorine elimination will furnish the corresponding six- and seven-membered ring products, respectively.

The structure of the final products is related to the nature of the electrophilic trapping reagents. The primary radical **G** may react preferentially because it is formed first and is more kinetically accessible. The tertiary radicals **D** and **H**, although more crowded, are more stable and survive longer in the reaction system. When reacting with ArBr via an oxidative addition/reductive elimination mechanism, primary radicals **C** and **G** are preferred because the process is more sensitive to steric hindrance of the ligand. More nucleophilic tertiary radicals **D** and **H** are favored when reacting with electron-deficient alkenes such as trifluoromethylalkenes or acrylates via Giese addition.

## Methods

### General procedure for the synthesis of 7-membered ring products

An oven-dried sealed tube equipped with a PTFE-coated stir bar was charged with NiBr$_2$(DME) (10 mol%), **L4** (20 mol%), trifluoromethyl alkene (0.1 mmol), 1,6-enyne (0.2 mmol), CsF (1.0 equiv), Mn powder (3.0 equiv), aryl bromide (2.0 equiv) and anhydrous DMA (2 mL). The sealed tube was sealed and removed from the glovebox.

Then the reaction was stirred at 60 °C until the reaction was complete (monitored by TLC). The resulting mixture was quenched with saturated NH$_4$Cl solution (5 mL) and further diluted with water (10 mL). The aqueous layer was extracted with EtOAc and the combined organic layers were washed with brine, dried over anhydrous Na$_2$SO$_4$, filtered, and concentrated under vacuum. The residue was purified by chromatography on silica gel, eluting with PE/EtOAc (50/1 ~ 5/1) to afford the desired 7-membered ring products.

### General procedure for the synthesis of 6-membered ring products

An oven-dried sealed tube equipped with a PTFE-coated stir bar was charged with NiBr$_2$(DME) (10 mol%), **L9** (20 mol%), trifluoromethyl alkene (0.1 mmol), 1,6-enyne (0.2 mmol), Mn powder (3.0 equiv), aryl bromide (2.0 equiv) and anhydrous DMA (2 mL). The sealed tube was sealed and removed from the glovebox. Then the reaction was stirred at 60 °C until the reaction was complete (monitored by TLC). The resulting mixture was quenched with saturated NH$_4$Cl solution (5 mL) and further diluted with water (10 mL). The aqueous layer was extracted with EtOAc and the combined organic layers were washed with brine, dried over anhydrous Na$_2$SO$_4$, filtered, and concentrated under vacuum. The residue was purified by chromatography on silica gel, eluting with PE/EtOAc (50/1 ~ 5/1) to afford the desired 6-membered ring products.

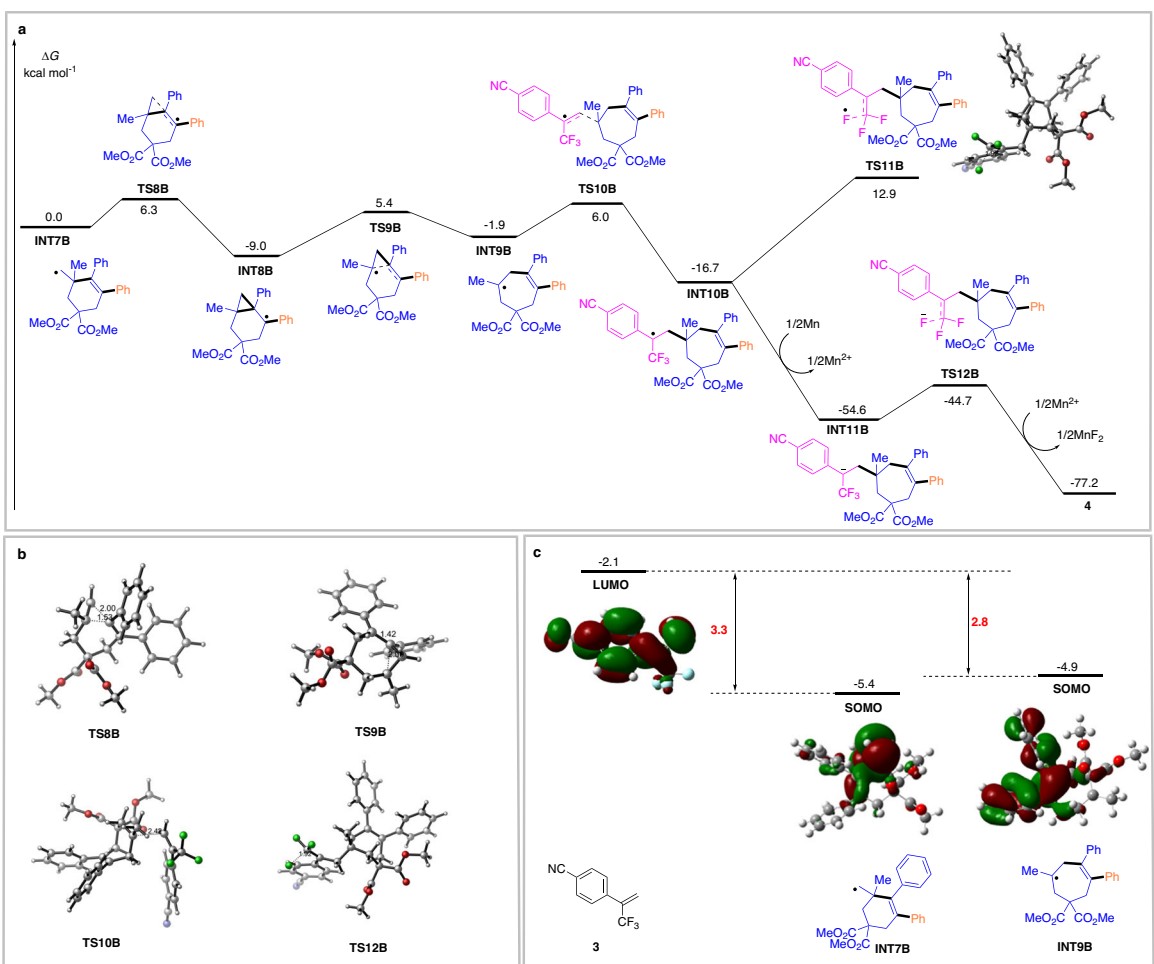

**Fig. 8 | The computational analysis of the radical INT7B. a** The calculated energy profile from radical **INT7B** to product **3**; **b** The structure of various transition states; **c** The frontier molecular orbital theory is further used to analysis the reactivity of the generated radicals.

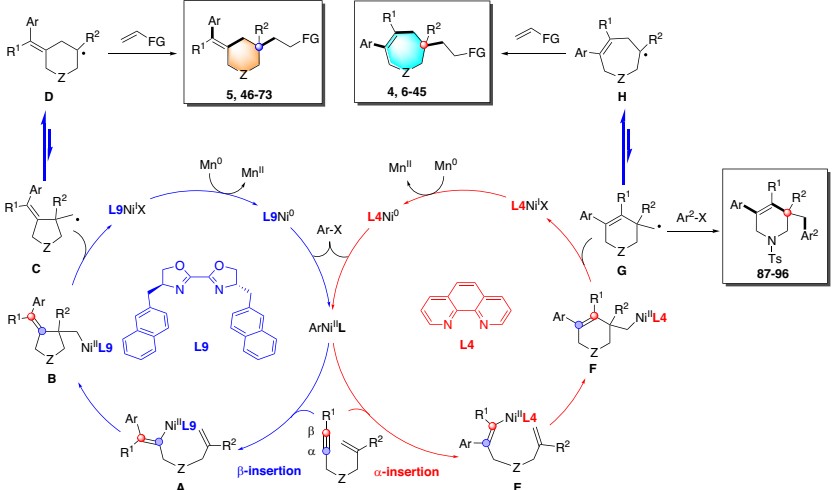

**Fig. 9 | Proposed mechanism.** Rationally explain the ligand-controlled regiodivergent arylation of alkynes and cyclization processes.

## Data availability

The authors declare that all the data supporting the findings of this work are available within the article and its Supplementary Information files or from the corresponding author upon request. The X-ray crystallographic coordinates for structures reported in this study have been deposited at the Cambridge Crystallographic Data Centre (CCDC), under deposition numbers 2220556 (**7**), 2220555 (**46**), and 2240774 (**74**). CIF files and crystal structure data are available in the Supplementary Information and Supplementary Data 1.

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

## Acknowledgements

Financial support provided by the National Natural Science Foundation of China (22171215 and 22171134), the Hubei Provincial Outstanding Youth Fund (2022CFA092), the National Key R&D Program of China (2022YFA1503200), and GuangDong Basic and Applied Basic Research Foundation (2022A1515010246). The authors also thank the High-Performance Computing Center of Nanjing University for doing the numerical calculations in this paper on its blade cluster system.

## Author contributions

W.K. and WF.L. conceived and designed the project. WF.L. and W.L. conducted the experiments. W.K., WF.L. and W.L. analyzed and interpreted the experimental data. M.W. and W.X. performed the DFT calculations. W.K. and M.W. prepared the manuscript. WF.L. and W.L. prepared the Supplementary Information. All authors contributed to discussions.

## Competing interests

The authors declare no competing interests.
