## [Peer Review File · Nature Communications]

REVIEWER COMMENTS

Reviewer #1 (Remarks to the Author):

This paper describes a ligand-controlled, nickel-catalyzed, switchable arylation cyclization of 1,6-enynes, with resulting trapping with trifluoromethyl-substituted alkenes, or further reaction with the aryl bromide to give various carbo- and heterocycles. Although the products are rather specialized, the scope of the reactions have been very well-explored, and the ability to access different scaffolds from a common precursor by changing the ligand is appealing (and mechanistically interesting). DFT studies were conducted, and a mechanism proposed that includes an interesting Beckwith-Dowd-like radical ring expansion. I think this paper is suitable for publication in Nature Communications after addressing several points:

1. The introduction appears to neglect mention of cyclizations of enynes that proceed by initial oxidative cyclization to form metallacyclic intermediates. This is a large class of reactions that proceed via a different mechanism to the ones presented in this paper (i.e. prior functionalization of the alkyne before cyclization of an alkenylmetal species). For completeness, this should be mentioned so that the reader is aware that this class of reaction is not being discussed.
2. In Scheme 1A, there appears to be a mistake in the picture of the endo-cyclization mode for beta-insertion (2nd reaction from top) as the resulting organometallic species has an extra methylene (CH₂) group that should not be there. The same applies to the product. Also, in the limitations box, it should say “regioselectivity in arylation of alkynes” rather than “regiospecificity in arylation of alkynes”. Even extremely high regioselectivity is not the same as regiospecificity, and I think “regioselective” should be used rather than “regiospecific”. In Scheme 1B, the ester next to the difluoromethylene group has a formatting error (overlaps with the structure below).
3. The first time Table 1, entry 1 is described, the text should mention that CsF was used. (It is only mentioned in the following sentence, which is a bit late.)
4. In Scheme 2, the equivalents of substrates used should be indicated – preferably in the generic reaction at the top (rather than in the footnotes).
5. In terms of the reaction scope in the paper, it would be useful to know whether heteroaromatics containing basic nitrogen atoms are tolerated. There is one nice example of this (product 41) but do nitrogen-containing heteroaryl bromides that are not indoles/carbazoles work (e.g. bromopyridines) work?

What happens when the quinoline-substituted trifluoromethyl alkene used to make product 41 is used in Scheme 3 and 4?

6. Regarding ligand 9, the Supporting Information does not appear to show how this was made and no characterization data are provided. These points should be addressed. Is this compound racemic

or enantiopure? In Schemes 3, 4, and 11, the stereochemistry of ligand 9 should be indicated (even if it is a racemic compound), because for a compound with two stereocenters, two diastereomers are possible.

7. For products 67, 68, 70, and 71 (and to a minimal extent, product 69), I would just like to check that the calculations for the yields of the reactions have taken into account the presence and ratio/mount of the reductive Heck product that is present in the product mixtures (as they have different molecular masses).

8. Regarding the DFT calculations and possible reaction mechanism, I am not an expert in computational chemistry by any means, but several recent papers have suggested E/Z isomerization of alkenylnickel species does not take place via carbene-like intermediates, but through η^2 -coordinated species. See:

<https://pubs.rsc.org/en/content/articlelanding/2023/qo/d3qo00723e>

<https://pubs.rsc.org/en/content/articlelanding/2023/qo/d3qo00701d>

<https://pubs.acs.org/doi/full/10.1021/jacs.2c03777>

I feel this possibility should be considered/computed by the authors and if required, appropriate changes made to the manuscript.

9. In the Supporting Information, large tables of computed energies and coordinates are not all that useful for the reader. I would encourage the authors to make this data available as an attachment (either as Supporting Information or in a data repository) that can be downloaded by readers.

Reviewer #2 (Remarks to the Author):

Diverse synthesis of cyclic compounds from the same starting materials is of great interest and significance in the synthetic chemistry, while very challenging. In this article, Kong and co-workers have developed a nickel-catalyzed ligand-controlled switchable arylation/cyclization cascade reactions involving 1,6-enynes with different electrophiles under reductive condition. The authors demonstrate the steric repulsion between the ligand and phenyl unit of alkyne dictates the regioselectivity of alkyne arylation, while the electrophilic trapping reagents determine the selectivity of the cyclization process. The reactions show broad substrate scope and generally good yields. The combination of experimental and DFT studies reveal that the reaction mechanism involves trans-/cis- arylation of alkyne controlled by ligand, exo-cyclization, followed by Dowd-Beckwith-type ring expansion to form endo-cyclization products. This work indeed provides an elegant strategy for obtaining the uncommon endo-cyclization products. On the whole, this is a very nice work and the reviewer would like to recommend it for publication in Nature Communications.

Several points as below could be considered or explained to make this manuscript much more suitable for publication.

- a. Did the authors observe the side products by direct reductive coupling of aryl bromides with CF₃-alkenes in current transformations and if so, how to suppress it? (author's previous report: *Chin. J. Chem.* 2022, 40, 2212)?
- b. What happens when L9 acts as a ligand via β -insertion without electron-deficient alkenes? The diarylated cyclization of 1,6-enynes?
- c. Considering the use of chiral L9 in cis-arylation cyclization, were ee values observed? And, stereoconfiguration of L9 in reaction equation of Scheme 3 should be redrawn.
- d. Main reaction conditions should be given in every Scheme (scope), especially for Scheme 5 (diarylated cyclization, ratio of two reactants). That is helpful for readers.
- e. The format of references should be checked again, keeping unified, such as refs.7-11.

Reviewer #3 (Remarks to the Author):

The paper describes a nickel-catalysed, ligand-controlled switchable arylation/cyclization of 1,6-enynes to produce piperidines and azepanes. Additionally, experimental and density functional theory studies get insight into the nature of regioselectivity for the insertion process of alkyne. This work achieved some interesting extension on the topic of switchable or divergent cyclization cascades of methacryloyl *o*-bromobenzamide and analogues with trifluoromethyl alkenes (*JACS*, 2022, 144,11626, *Angew Chem*, 2022, 61, e202201574 from these author's own group).

Undoubtedly, the most intriguing and potentially impactful part of the manuscript is the mechanism. The authors have evidence for regioselectivity of the alkyne arylation process that may arise from the controlling effects of steric hindrance of the ligand and claim that regioselectivity of the insertion process of alkyne was entirely controllable. However, the 1,6-enynes substrates described in this manuscript were limited to these ones with pendant steric arylalkyne motifs. Therefore, these authors should further investigate the regioselectivity of the insertion process of other alkynes with small terminal substituents (e.g., Me, Et) and even terminal alkynes.

The yields are acceptable, but the reaction is mainly limited to the use of α -trifluoromethyl alkenes as the acceptors. Are other activated alkenes, as vinyl sulfones, compatible with the reported reaction?

The trans-arylate exo-cyclization/cross-coupling to give diarylated products was also studied in the manuscript. Being different from using α -trifluoromethyl alkenes as the acceptor, the Dowd-Beckwith ring expansion seem not to be observed. Can these authors give reasonable explanation for this regard.

Ligand-controlled arylate cyclization for the divergent synthesis of chiral piperidines and azepanes was studied, enabled by using pre-prepared enantioenriched 1,6-enyne. It seems more likely to the reviewer that this diastereoselectivity is due to the pattern driven by stable-conformation effects of six-membered cycles. But some references on related topic seem not to be cited in this manuscript (Nature Chemistry 2010, 2, 125-130, and etc.)

In addition, there are a few corrections to be addressed, a) in Table 1, the L 9 was showed as a chiral ligand, but the same Ligand in Scheme 3 and Scheme 4 was showed as an achiral ligand; b) in scheme (B), the CO₂R₂ group was not showed in correct place. The text, in general, should be re-checked for minor typo corrections.

In summary, in my opinion this manuscript would be suitable for publication in NC, but substantial changes need to be implemented (major revision) before acceptance to publication.

This manuscript presents a nickel-catalyzed switchable arylation/endo-cyclization of 1,6-enynes which can lead to *endo*-cyclization products. Based on extensive experimental and computational results, the authors proposed the reaction mechanism involves *exo*-cyclization followed by Dowd-Beckwith-type ring expansion, other than through a retro 6-*exo* cyclization via β -carbon elimination followed by 7-*endo* cyclization mechanism, or 1,2-dyotropic rearrangement mechanism. In addition, it's revealed the regioselectivity of the alkyne arylation process is entirely controlled by the steric hindrance of the ligand, while the electrophilic trapping reagents determine the selectivity of the cyclization mode.

I think the work is of significance in both academic and practical perspectives. However, there are still some defects which have reduced its readability and need further improvements.

(1) In both the main text and the Supplementary Information, some necessary guidelines for figures are missed, thus the reader needs to spend much time on seeking for the right figure in order to find specific contents or structures.

For example,

Line 5, page 15, "while the influence of ligand L9 is shown in Supplementary Information." If "(**Fig. S6**)" is added into the sentence, to make it read as "while the influence of ligand L9 is shown in Supplementary Information (**Fig. S6**).", we can find it very fast, and don't need to browse the entire SI just for the sake of finding the corresponding **Fig. S6**.

Line 2, page 17, "We located a homolytic cleavage pathway of alkyl-nickel(II) to generate radical **INT7B**." I think it's better to add "(**Scheme 10A**)" and make it read as "We located a homolytic cleavage pathway of alkyl-nickel(II) to generate radical **INT7B** (**Scheme 10A**)."

Similarly, line 10-11, page 18, "(for the details of the computational data, see **Figure S8** in the Supplementary Information)"

(2) Line 6, page 18, "Owing to the computed high energy barrier (**TS11B**, $\Delta G^\ddagger = 46.3$ kcal/mol).....", I found **TS11B** only in **Scheme 10A** in the main text, and don't understand quite well, how an energy barrier of 46.3 kcal/mol is calculated accordingly?

(3) In **Scheme 8A**, could any transition state corresponding to the C=C double bond rotation from **INT4B-L4** to **INT5B-L4** be located?

(4) In **Scheme 8B**, how about conducting NCI (noncovalent interaction) analysis to visualize the steric interaction between substrates and ligands, for example, in **TS4A-L4**, **TS4B-L4**, **TS4A-L9** and **TS4B-L9**?

(5) For references 7-11, their formats differ significantly from all others.

(6) In the Supplementary Information, Page S22, “The calculated results indicated the geometry optimizations conducted in gas phase have little influence on the energies of transition states.” Should it be “The calculated results indicated the geometry optimizations conducted in gas phase have **little** influence on the **free energy barriers** of transition states.”?

Page S23, there are more than one representation for ω B97X-D, for example, “ $\omega\beta 97x$ -D” (Line 1), “ $\omega b97xd$ ” (Table S2). These simple errors should be avoided.

In Table S2, could any instructions for the calculation of G_{corr} (from the original Gaussian out/log files) be given?

Page S27, in the caption of **Figure S4**, “(B3LYP-D3/6-31G(d)-LANL08)” seems a note, maybe a complete sentence illustrating the computational level is preferable.

Page S27 and S28, the author says “The CF_3 group increases the electrophilic character of **INT10B**, which should be thermodynamically more favorable to recombine with intermediate **INT6B-L4** to generate Ni(II) complex **INT11C**.”, “we suspect that the recombination of **INT6B-L4** with radical **INT10B** is a reversible process”, “The corresponding results have been added in Section 6 of the Supporting Information.”. However, I couldn’t find the energetic profiles corresponding to the combination of **INT6B-L4** with **INT10B** to generate **INT11C** in the Supporting Information, and am unable to determine whether it is reversible or not.

Page S28, “If **INT11C** was formed, the energy barrier of the following β -F elimination was 27.0 kcal mol⁻¹, which was much higher than that of reductive process from **INT6B-L4** to **INT1A-L4** (**Figure S5**).”. Since **INT6B-L4** and **INT1A-L4** are not presented in **Figure S5**, I think it’s misleading when “(**Figure S5**)” is placed as it is. It should be “If **INT11C** was formed, the energy barrier of the following β -F elimination was 27.0 kcal mol⁻¹ (**Figure S5**), which was much higher than that of reductive process from **INT6B-L4** to **INT1A-L4** (**Scheme 8A**).”

(7) In addition, the “kcal mol⁻¹”, “kcal/mol” (“kcal/mol” in the main text, but “kcal mol⁻¹” in figures and the Supporting Information) should be unified, including “Fig.” and “Figure” in the Supporting Information (**Fig. S6** is in abbreviation, while all others are not abbreviated). The unit for energies in **Figures S1** and **S2** are missed. The font size and structures in some figures (for example, in **Figures S6** and **S8**) look obviously smaller than in other ones.

... ..

In a word, I think the results presented in this manuscript are pretty good, but the writing seems a bit coarse. The authors should check their manuscript more carefully and meet all points listed above in the revised version. Then it can meet the criterion of *Nature Communications* and be considered for publication on it.

Response to Reviewers

Reviewer #1 (Remarks to the Author):

This paper describes a ligand-controlled, nickel-catalyzed, switchable arylation cyclization of 1,6-enynes, with resulting trapping with trifluoromethyl-substituted alkenes, or further reaction with the aryl bromide to give various carbo- and heterocycles. Although the products are rather specialized, the scope of the reactions have been very well-explored, and the ability to access different scaffolds from a common precursor by changing the ligand is appealing (and mechanistically interesting). DFT studies were conducted, and a mechanism proposed that includes an interesting Beckwith-Dowd-like radical ring expansion. I think this paper is suitable for publication in Nature Communications after addressing several points:

(1) *The introduction appears to neglect mention of cyclizations of enynes that proceed by initial oxidative cyclization to form metalacyclic intermediates. This is a large class of reactions that proceed via a different mechanism to the ones presented in this paper (i.e. prior functionalization of the alkyne before cyclization of an alkenyl metal species). For completeness, this should be mentioned so that the reader is aware that this class of reaction is not being discussed.*

Answer: Thanks to the reviewers for their very positive comments.

Transition-metal-catalyzed cycloisomerization⁷⁻¹¹ and reductive cyclization¹²⁻¹⁶ of 1,6-enynes have been established to generate five-membered (hetero)cycles, and such reactions often proceed via oxidative cyclization.

To make it clearer, we have added the following sentences to the revised manuscript: “*From a reaction mechanism perspective, these transformations are primarily initiated by the formation of metalacyclic intermediates via oxidative cyclization pathways or the formation of alkenyl metal species via carbometallation processes. The former has been well explored,⁷⁻¹⁶ but controlling the regioselectivity of the latter is difficult, usually in two ways: α -insertion or β -insertion (Scheme, 1A).*”

(2) *In Scheme 1A, there appears to be a mistake in the picture of the endo-cyclization mode for beta-*

insertion (2nd reaction from top) as the resulting organometallic species has an extra methylene (CH₂) group that should not be there. The same applies to the product. Also, in the limitations box, it should say “regioselectivity in arylation of alkynes” rather than “regiospecificity in arylation of alkynes”. Even extremely high regioselectivity is not the same as regiospecificity, and I think “regioselective” should be used rather than “regiospecific”. In Scheme 1B, the ester next to the difluoromethylene group has a formatting error (overlaps with the structure below).

Answer: Thank you for your patience. All these mistakes have been corrected in the revised manuscript.

(3) The first time Table 1, entry 1 is described, the text should mention that CsF was used. (It is only mentioned in the following sentence, which is a bit late.)

Answer: We have added a description of using CsF in the main text.

(4) In Scheme 2, the equivalents of substrates used should be indicated – preferably in the generic reaction at the top (rather than in the footnotes).

Answer: As suggested, the equivalents of substrates used are indicated in the equations of the revised manuscript.

(5) In terms of the reaction scope in the paper, it would be useful to know whether heteroaromatics containing basic nitrogen atoms are tolerated. There is one nice example of this (product 41) but do nitrogen-containing heteroaryl bromides that are not indoles/carbazoles work (e.g. bromopyridines) work? What happens when the quinoline-substituted trifluoromethyl alkene used to make product 41 is used in Scheme 3 and 4?

Answer: We performed the reaction using nitrogen-containing heteroaryl bromide such as 5-bromo-2-methoxypyridine, the trans-arylation/endo-cyclization product was obtained in 67% yield.

Quinoline-substituted trifluoromethyl alkene undergoes cis-arylation/endo cyclization to give the corresponding product in 59% yield. Unfortunately, quinoline-substituted trifluoromethyl alkene was found to be incompatible with diastereoselective cyclization, and the desired product was not observed. All these results were added in the revised manuscript.

(6) Regarding ligand 9, the Supporting Information does not appear to show how this was made and no characterization data are provided. These points should be addressed. Is this compound racemic or enantiopure? In Schemes 3, 4, and 11, the stereochemistry of ligand 9 should be indicated (even if it is a racemic compound), because for a compound with two stereocenters, two diastereomers are possible.

Answer: L9 is a known compound and we synthesized it according to the reference (*Tetrahedron: Asymmetry*, **2014**, 25, 936-943). L9 is enantiopure and its absolute configuration is indicated in the revised manuscript.

(7) For products 67, 68, 70, and 71 (and to a minimal extent, product 69), I would just like to check that the calculations for the yields of the reactions have taken into account the presence and ratio/mount of the reductive Heck product that is present in the product mixtures (as they have different molecular masses).

Answer: Thank you for your patience. We have recalculated the yields of the above-mentioned products and confirmed that these yields are accurate.

(8) Regarding the DFT calculations and possible reaction mechanism, I am not an expert in computational chemistry by any means, but several recent papers have suggested E/Z isomerization of alkenylnickel species does not take place via carbene-like intermediates, but through eta-2-coordinated species. See: <https://pubs.rsc.org/en/content/articlelanding/2023/qo/d3qo00723e>

<https://pubs.rsc.org/en/content/articlelanding/2023/qa/d3qa00701d>

<https://pubs.acs.org/doi/full/10.1021/jacs.2c03777>

I feel this possibility should be considered/computed by the authors and if required, appropriate changes made to the manuscript.

Answer: According to previous mechanistic studies of transition metal-catalyzed alkyne functionalization, two possible modes of alkenyl metal *Z/E* isomerization are summarized. We considered both possibilities in DFT calculations.

The charge-separated resonance structure, which is 9.2 kcal mol⁻¹ higher in energy than the intermediate **INT4B-L4**, undergoes single-bond rotation with an energy barrier of 4.2 kcal mol⁻¹. This results in a total energy of 13.4 kcal mol⁻¹ for the direct unimolecular rotation. Alternatively, the intermediate **INT4B-L4** undergoes reversible Ni–C bond homolysis through the transition state **TS5B-L4-1** with an energy barrier of 11.0 kcal mol⁻¹. As a result, a vinyl radical, **INT4B-L4-radical**, is generated. This carbon-centered radical adopts a linear geometry, which then undergoes radical recombination with the Ni(I) center, leading to the formation of the more stable intermediate **INT5B-L4**. The computational results support the mechanism involving reversible bond homolysis.

(9) *In the Supporting Information, large tables of computed energies and coordinates are not all*

that useful for the reader. I would encourage the authors to make this data available as an attachment (either as Supporting Information or in a data repository) that can be downloaded by readers.

Answer: Thanks for your suggestion. Although computed energies and coordinates may not be useful to all readers, they are necessary supporting data for theoretical calculations. Therefore, we still keep these data in the Supporting Information to facilitate access to these data for readers interested in DFT calculations.

Reviewer #2 (Remarks to the Author):

*Diverse synthesis of cyclic compounds from the same starting materials is of great interest and significance in the synthetic chemistry, while very challenging. In this article, Kong and co-workers have developed a nickel-catalyzed ligand-controlled switchable arylation/cyclization cascade reactions involving 1,6-enynes with different electrophiles under reductive condition. The authors demonstrate the steric repulsion between the ligand and phenyl unit of alkyne dictates the regioselectivity of alkyne arylation, while the electrophilic trapping reagents determine the selectivity of the cyclization process. The reactions show broad substrate scope and generally good yields. The combination of experimental and DFT studies reveal that the reaction mechanism involves trans-/cis- arylation of alkyne controlled by ligand, exo-cyclization, followed by Dowd-Beckwith-type ring expansion to form endo-cyclization products. This work indeed provides an elegant strategy for obtaining the uncommon endo-cyclization products. **On the whole, this is a very nice work and the reviewer would like to recommend it for publication in Nature Communications.***

Several points as below could be considered or explained to make this manuscript much more suitable for publication.

(1) Did the authors observe the side products by direct reductive coupling of aryl bromides with CF₃-alkenes in current transformations and if so, how to suppress it? (author's previous report: Chin. J. Chem. 2022, 40, 2212)?

Answer: This is a good question. In fact, direct cross-coupling of aryl halides with trifluoromethyl

alkenes is known, however, under our standard conditions, only trace amounts of direct reductive coupling products of aryl bromides with trifluoromethyl alkenes were detected.

We believe that the possible reason is that the carbon-carbon triple bonds in the 1,6-enyne is more reactive, which reacts preferentially with arylnickel species rather than with trifluoromethyl alkenes.

(2) *What happens when L9 acts as a ligand via β -insertion without electron-deficient alkenes? The diarylated cyclization of 1,6-enynes?*

Answer: Using **L9** as a ligand, we performed the reaction of 1,6-enyne with phenyl bromide in the absence of trifluoromethyl alkene. However, no expected diarylation product was detected, and only 49% of the by-product of 1,6-enyne decomposition was isolated. The possible reason is that sterically hindered **L9** may hinder the cross-coupling of σ -alkyl-Ni intermediates and aryl bromides.

(3) *Considering the use of chiral L9 in cis-arylation cyclization, were ee values observed? And, stereoconfiguration of L9 in reaction equation of Scheme 3 should be redrawn.*

Answer: Despite the use of the chiral Box ligand **L9** in the *cis*-arylation cyclization, only 4% ee was observed and these results are summarized in Table S1.

In the revised manuscript, we have redrawn the absolute configuration of ligand **L9**.

(4) *Main reaction conditions should be given in every Scheme (scope), especially for Scheme 5 (diarylated cyclization, ratio of two reactants). That is helpful for readers.*

Answer: We have added reaction conditions to the footnotes of each Schemes in the revised manuscript.

(5) *The format of references should be checked again, keeping unified, such as refs.7-11.*

Answer: Thanks to the reviewers' patience, we rechecked the references and corrected the format of the above references in the revised manuscript.

Reviewer #3 (Remarks to the Author):

The paper describes a nickel-catalysed, ligand-controlled switchable arylation/cyclization of 1,6-enynes to produce piperidines and azepanes. Additionally, experimental and density functional theory studies get insight into the nature of regioselectivity for the insertion process of alkyne. This work achieved some interesting extension on the topic of switchable or divergent cyclization cascades of methacryloyl o-bromobenzamide and analogues with trifluoromethyl alkenes (JACS, 2022, 144,11626, Angew Chem, 2022, 61, e202201574 from these author's own group).

Undoubtedly, the most intriguing and potentially impactful part of the manuscript is the mechanism. The authors have evidence for regioselectivity of the alkyne arylation process that may arise from the controlling effects of steric hindrance of the ligand and claim that regioselectivity of the insertion process of alkyne was entirely controllable.

(1) However, the 1,6-enynes substrates described in this manuscript were limited to these ones with pendant steric arylalkyne motifs. Therefore, these authors should further investigate the regioselectivity of the insertion process of other alkynes with small terminal substitutes (e.g., Me, Et) and even terminal alkynes.

Answer: Thank you for your suggestion. We tested methyl-substituted and terminal alkyne-substituted 1,6-enyne substrates and, unfortunately, no target product was detected. These results indicate that aryl substitution of the alkyne moiety is essential in our reaction. There may be two reasons. On the one hand, the aryl group can activate the carbon-carbon triple bond, so the aryl nickel species reacts with the activated alkyne first, rather than with the trifluoromethyl alkene; on the other hand, the presence of the aryl group helps to stabilize the formed alkenyl nickel intermediate .

But for aryl-substituted 1,6-enynes, products with different scaffolds can be obtained from the same starting material by simply changing the ligands. It is clear that the regioselectivity of the alkyne arylation process is controlled by the ligand rather than the substrate, which is beyond doubt.

endo-selective cyclization

Different types of 1,6-enyne substrates:

Lautens reported the use of ammonium halides as HX surrogates to accomplish a Pd-catalyzed hydrohalogenation of aryl-substituted 1,6-enynes as shown below. **The reaction relies on the use of *tert*-butoxy as a nitrogen protecting group, giving a 4.2:1 ratio of *exo/endo* cyclized products.**

Figure 1. Pd-catalyzed hydroiodination of 1,6-enynes using Et₃N-HI as an HI surrogate: Effect of the N-protecting group. Notes: Reactions were run on a 0.2 mmol scale and yields determined by ¹H NMR analysis of the crude reaction mixture; values in parentheses represent the isolated yield of 2, and the ratios in square brackets represent yields in the form of [2:(3 + 4)]. Superscript indicates that subsection of 1k to the reaction conditions led to formation of 1b in 64% yield. ND = not determined.

Table 1. Pd-Catalyzed Hydroiodination of 1,6-Enynes: Effect of Reaction Parameters^{a†}

entry	variation from the "standard" conditions	yield ^{b,c} 2a (%)	yield ^b 3a + 4a (%)
1	none	75 (72)	18
2	no Pd(P ^t Bu) ₃ ₂	0	0
3	1,4-dioxane instead of PhMe	75	21
4	110 °C instead of 120 °C	61	15
5	100 °C instead of 120 °C	56	13
6	0.1 M instead of 0.05 M	72	18
7	Pd(QPhos) ₂ instead of Pd(P ^t Bu) ₃ ₂	30	15
8 ^{d†}	P(1-Ad) ₃ /Pd ₂ (dba) ₃ instead of Pd(P ^t Bu) ₃ ₂	56	17
9 ^{d†}	DTBNpP/Pd ₂ (dba) ₃ instead of Pd(P ^t Bu) ₃ ₂	24	9
10	5 mol % Pd(P ^t Bu) ₃ ₂	65	19
11	1.0 equiv Et ₃ N-HI	68	17
12	2.0 equiv Et ₃ N-HI	69	17
13	Me ₃ N-HI instead of Et ₃ N-HI	37	8
14	Me ₃ N-HI/1,4-dioxane instead of Et ₃ N-HI/PhMe	66	15
15	^t Bu ₃ N-HI instead of Et ₃ N-HI	71	24

^aReactions were run on a 0.2 mmol scale. ^bDetermined by ¹H NMR analysis of the crude reaction mixture using 1,3,5-trimethoxybenzene as an internal standard. ^cValue in parentheses represents the isolated yield. ^d[Pd] = 10 mol %, phosphine = 20 mol %. 1-Ad = 1-adamantyl. DTBNpP = di(*tert*-butyl)neopentylphosphine.

J. Am. Chem. Soc. **2017**, *139*, 3546-3557.

A remarkable advance of our work is the highly selective *endo*-cyclization (both the six-membered and seven-membered ring products were generated by formal *endo*-selective cyclizations).

(2) The yields are acceptable, but the reaction is mainly limited to the use of α -trifluoromethyl alkenes as the acceptors. Are other activated alkenes, as vinyl sulfones, compatible with the reported reaction?

Answer: This transformation is not limited to trifluoromethyl alkenes, as it is also compatible with other electron-deficient olefins such as acrylate (44). Vinyl sulfone can also react smoothly to give the target product in 33% yield (45). And this result has been added in the revised manuscript.

(3) *The trans-arylativ exo-cyclization/cross-coupling to give diarylated products was also studied in the manuscript. Being different from using α -trifluoromethyl alkenes as the acceptor, the Dowd-Beckwith ring expansion seem not to be observed. Can these authors give reasonable explanation for this regard.*

Answer: This is a very good question. In fact, our research group has been expanding the “terminating” reagents for this type of reaction. So far, we have only found electron-deficient alkenes (trifluoromethyl alkenes, acrylates, and vinyl sulfinates) to be compatible.

Possible reasons are as follows:

Homolytic cleavage of the nickel alkyl intermediate **A** will generate the corresponding primary alkyl radical **B**. Further cyclization/opening will lead to a more stable tertiary radical intermediate **C**. The primary radical might be expected to react preferentially as it is kinetically more accessible. However, the tertiary radical intermediate **C** is more stable.

The key to the success of our strategy is the use of less sterically hindered and highly reactive electron-deficient alkenes (trifluoromethyl alkene) as electrophilic trapping agents, which reverse this natural tendency and preferentially react with tertiary radicals.

The fact that 6-*exo* cyclization/cross-coupling products were not detected is due to the irreversible Dowd-Beckwith-type ring expansion is kinetically more favorable than the reaction of trifluoromethyl alkenes with primary alkyl radical intermediate **B**. This result has been confirmed by DFT calculations.

(4) *Ligand-controlled arylation cyclization for the divergent synthesis of chiral piperidines and azepanes was studied, enabled by using pre-prepared enantioenriched 1,6-enyne. It seems more likely to the reviewer that this diastereoselectivity is due to the pattern driven by stable-conformation*

effects of six-membered cycles. But some references on related topic seem not to be cited in this manuscript (Nature Chemistry 2010, 2, 125-130, and etc.)

Answer: Thanks for your suggestion. We have cited the corresponding references (*Chem. Rev.* **96**, 835–876 (1996); *Nat. Chem.* **2**, 125-130 (2010)) in the revised manuscript.

(5) *In addition, there are a few corrections to be addressed, a) in Table 1, the L 9 was showed as a chiral ligand, but the same Ligand in Scheme 3 and Scheme 4 was showed as an achiral ligand; b) in scheme (B), the CO₂R₂ group was not showed in correct place. The text, in general, should be re-checked for minor typo corrections.*

Answer: All these errors have been corrected in the revised manuscript.

In summary, in my opinion this manuscript would be suitable for publication in NC, but substantial changes need to be implemented (major revision) before acceptance to publication.

Reviewer #4 (Remarks to the Author):

*This manuscript presents a nickel-catalyzed switchable arylation/endo-cyclization of 1,6-enynes which can lead to endo-cyclization products. Based on extensive experimental and computational results, the authors proposed the reaction mechanism involves exo-cyclization followed by Dowd-Beckwith-type ring expansion, other than through a retro 6-exo cyclization via β -carbon elimination followed by 7-endo cyclization mechanism, or 1,2-dyotropic rearrangement mechanism. In addition, it's revealed the regioselectivity of the alkyne arylation process is entirely controlled by the steric hindrance of the ligand, while the electrophilic trapping reagents determine the selectivity of the cyclization mode. **I think the work is of significance in both academic and practical perspectives.** However, there are still some defects which have reduced its readability and need further improvements.*

(1) *In both the main text and the Supplementary Information, some necessary guidelines for figures are missed, thus the reader needs to spend much time on seeking for the right figure in order to find specific contents or structures.*

For example, Line 5, page 15, “ while the influence of ligand L9 is shown in Supplementary Information” If “(Fig. S6)” is added into the sentence, to make it read as “while the influence of

ligand L9 is shown in Supplementary Information (Fig. S6).”, we can find it very fast, and don't need to browse the entire SI just for the sake of finding the corresponding Fig. S6.

Answer: Thanks for your suggestion. In order to improve the readability of the article, we have incorporated detailed location information from the Supplementary Information into the revised manuscript.

(2) *Line 2, page 17, “We located a homolytic cleavage pathway of alkyl-nickel(II) to generate radical INT7B.” I think it's better to add “(Scheme 10A)” and make it read as “We located a homolytic cleavage pathway of alkyl-nickel(II) to generate radical INT7B(Scheme10A).”*

Similarly, line 10-11, page 18, “(for the details of the computational data, see Figure S8 in the Supplementary Information)”

Answer: We have made revisions to the article.

(3) *Line 6, page 18, “Owing to the computed high energy barrier (TS11B, $\Delta G^\ddagger = 46.3 \text{ kcal/mol}$).....”, I found TS11B only in Scheme 10A in the main text, and don't understand quite well, how an energy barrier of 46.3 kcal/mol is calculated accordingly?*

Answer: We have revised this mistake. The energy barrier of transition state **TS11B** is 29.6 kcal mol⁻¹ relative to **INT10B** (the relative Gibbs free energy of **TS11B** 12.9 kcal·mol⁻¹ minus the energy of **INT10B** -16.7 kcal·mol⁻¹).

(4) *In Scheme 8A, could any transition state corresponding to the C=C double bond rotation from INT4B-L4 to INT5B-L4 be located?*

Answer: Further literature research reveals that the *cis/trans* tautomerization process involves multiple steps. During this process, the intermediate **INT4B-L4** undergoes reversible Ni–C bond homolysis through the transition state **TS5B-L4-1** with an energy barrier of 11.0 kcal mol⁻¹. As a result, a vinyl radical, **INT4B-L4-radical**, is generated. This carbon-centered radical adopts a linear geometry, which then undergoes radical recombination with the Ni(I) center, leading to the formation of the more stable intermediate **INT5B-L4**.

(5) In Scheme 8B, how about conducting NCI (noncovalent interaction) analysis to visualize the steric interaction between substrates and ligands, for example, in TS4A-L4, TS4B-L4, TS4A-L9 and TS4B-L9?

Answer: The NCI analysis between substrates and ligands in **TS4A-L4**, **TS4B-L4**, **TS4A-L9** and **TS4B-L9** has been conducted in the revised manuscript, and the results are shown in Figure 8B.

(6) For references 7-11, their formats differ significantly from all others.

Answer: we have corrected the formats of these references in the revised manuscript.

(7) In the Supplementary Information, Page S22, “The calculated results indicated the geometry optimizations conducted in gas phase have litter influence on the energies of transition states.” Should it be “The calculated results indicated the geometry optimizations conducted in gas phase have little influence on the free energy barriers of transition states.”?

Answer: We have made revisions to the Supplementary Information.

(8) Page S23, there are more than one representation for ωB97X-D, for example, “wβ97x-D”(Line 1), “ωb97xd” (Table S2). These simple errors should be avoided.

Answer: These mistakes has been revised.

(9) In Table S2, could any instructions for the calculation of Gcorr (from the original Gaussianout/log files) be given?

Answer: The input keywords for structure optimizations and solvation single-point energies have been illustrated in the footnote of Table S2.

(10) Page S27, in the caption of Figure S4, “(B3LYP-D3/6-31G(d)-LANL08)” seems a note, maybe a complete sentence illustrating the computational level is preferable.

Answer: We have used a complete sentence to describe the computational level in the caption of Figure S4.

(11) Page S27 and S28, the author says “The CF₃ group increases the electrophilic character of INT10B, which should be thermodynamically more favorable to recombine with intermediate INT6B-L4 to generate Ni(II) complex INT11C.”, “we suspect that there combination of INT6B-L4 with radical INT10B is a reversible process”, “The corresponding results have been added in Section 6 of the Supporting Information.”. However, I couldn’t find the energetic profiles corresponding to the combination of INT6B-L4 with INT10B to generate INT11C in the Supporting Information, and am unable to determine whether it is reversible or not.

Answer: The recombination of intermediate INT6B-L4 and radical INT10B is also calculated. The calculational results, demonstrating that this process is reversible, have been included in the supplementary information (Figure S5).

(12) Page S28, “If INT11C was formed, the energy barrier of the following β -F elimination was 27.0 kcal mol⁻¹, which was much higher than that of reductive process from INT6B-L4 to INT1A-L4

(Figure S5).". Since INT6B-L4 and INT1A-L4 are not presented in Figure S5, I think it's misleading when "(Figure S5)" is placed as it is. It should be "If INT11C was formed, the energy barrier of the following β -F elimination was 27.0 kcal mol⁻¹ (Figure S5), which was much higher than that of reductive process from INT6B-L4 to INT1A-L4 (Scheme 8A)."

Answer: thanks for constructive suggestion, we have made this change in the revised SI.

(13) In addition, the "kcal mol⁻¹", "kcal/mol" ("kcal/mol" in the main text, but "kcal mol⁻¹" in figures and the Supporting Information) should be unified, including "Fig." and "Figure" in the Supporting Information (Fig. S6 is in abbreviation, while all others are not abbreviated).

Answer: we have corrected these errors in the revised manuscript and SI.

(14) The unit for energies in Figures S1 and S2 are missed.

Answer: we have added the unit in Figures S1 and S2 in the revised SI.

(15) The font size and structures in some figures (for example, in Figures S6 and S8) look obviously smaller than in other ones. In a word,

Answer: We have adjusted the figures S6 and S8 in the Supplementary Information.

In a word, I think the results presented in this manuscript are pretty good, but the writing seems a bit coarse. The authors should check their manuscript more carefully and meet all points listed above in the revised version. Then it can meet the criterion of Nature Communications and be considered for publication on it.

REVIEWERS' COMMENTS

Reviewer #1 (Remarks to the Author):

The authors have appeared to address most of the points raised by the reviewers satisfactorily (though I cannot judge the detailed computational chemistry points), which further strengthens the case for this nice work to be published in Nature Communications. However, there are still some points remaining that should improve the work.

1. The authors have looked at the computational work again and they proposed a revised mechanism for alkenylnickel E/Z isomerization that involves reversible homolysis of the carbon-nickel bond to generate an alkenyl radical, which can undergo the isomerization before recombination with the nickel. This is opposed to the original proposal that invokes nickel carbene-like intermediates. If the computational work supports this, then fine. However, the authors have not appeared to address satisfactorily the original reviewer's comment that other groups have proposed alkenylnickel E/Z isomerization in related reactions proceeds via eta-2-coordinated alkenylnickel species, rather than through reversible homolysis of the carbon-nickel bond. Three relevant papers are below:

<https://pubs.rsc.org/en/content/articlelanding/2023/qo/d3qo00723e>

<https://pubs.rsc.org/en/content/articlelanding/2023/qo/d3qo00701d>

<https://pubs.acs.org/doi/full/10.1021/jacs.2c03777>

At a bare minimum, these papers should be cited and there should be discussion in the paper about what these other groups proposed, and how that is different from the proposal of reversible homolysis of the carbon-nickel bond by the authors. Ideally, the authors should briefly examine this alternative mechanism for E/Z isomerization (involve eta-2-coordinated species) computationally, to see how that compares, and why that would not operate in the reactions described in this paper.

2. There is one instance of "cis/trans tautomerization", which needs correcting to "cis/trans isomerization".

3. Regarding the large tables of computed energies and coordinates in the Supporting Information, the suggestion was not to actually remove these from the SI, but rather, to encourage the authors to make the raw data available as an attachment (either as Supporting Information or in a data repository) that can be downloaded by readers, and then examined using computational chemistry software. Apologies if this was not 100% clear.

4. The authors have satisfactorily answered most of the questions from the reviewers regarding substrate scope (what works and what doesn't work), and about what happens when certain suggested experiments are carried out. However, not all of this information is provided in the paper (or SI) itself. I am not suggesting that the authors provide full explanations in the paper, i.e.,

reproducing exactly what is in the rebuttal letter, but for certain cases, I think it would be useful to add a brief comment in the paper to explain. E.g. for points 1 and 2 of Reviewer 2, and point 1 of Reviewer 3.

Reviewer #2 (Remarks to the Author):

This revised manuscript has addressed all issues raised by all of the reviewers, and thus is recommended to be accepted for publication without further revisions.

Reviewer #3 (Remarks to the Author):

Kong and co-workers have tried best to fulfill the referees' suggestions on this manuscript. Thus, I suggest its acceptance for publication in nature communications.

This paper is a revised manuscript previously submitted to *Nature Communications* as a full paper. The authors have added necessary computational trials to address all my concerns in my initial comments, although there's a trivial mistake in their rebuttal ("Figure 8B" on page 14 should be Scheme 8B). Now the computational results are more detailed and reliable, and the writing looks more readable. I think the present manuscript is suitable for publication on *Nature Communications* as it stands.

Response to Reviewers

Reviewer #1 (Remarks to the Author):

The authors have appeared to address most of the points raised by the reviewers satisfactorily (though I cannot judge the detailed computational chemistry points), which further strengthens the case for this nice work to be published in Nature Communications.

However, there are still some points remaining that should improve the work.

1. *The authors have looked at the computational work again and they proposed a revised mechanism for alkenylnickel E/Z isomerization that involves reversible homolysis of the carbon-nickel bond to generate an alkenyl radical, which can undergo the isomerization before recombination with the nickel. This is opposed to the original proposal that invokes nickel carbene-like intermediates. If the computational work supports this, then fine. However, the authors have not appeared to address satisfactorily the original reviewer's comment that other groups have proposed alkenylnickel E/Z isomerization in related reactions proceeds via eta-2-coordinated alkenylnickel species, rather than through reversible homolysis of the carbon-nickel bond. Three relevant papers are below:*

<https://pubs.rsc.org/en/content/articlelanding/2023/qo/d3qo00723e>

<https://pubs.rsc.org/en/content/articlelanding/2023/qo/d3qo00701d>

<https://pubs.acs.org/doi/full/10.1021/jacs.2c03777>

At a bare minimum, these papers should be cited and there should be discussion in the paper about what these other groups proposed, and how that is different from the proposal of reversible homolysis of the carbon-nickel bond by the authors. Ideally, the authors should briefly examine this alternative mechanism for E/Z isomerization (involve eta-2-coordinated species) computationally, to see how that compares, and why that would not operate in the reactions described in this paper.

Answer: Thanks for your suggestion. We have also considered the mechanism that alkenylnickel E/Z isomerization in related reactions proceeds via eta-2-coordinated alkenylnickel species. The energy barrier of **TS5C-L4** is 10.4 kcal mol⁻¹, which is comparable to that of **TS5B-L4-1** (10.4 kcal mol⁻¹ vs 10.9 kcal mol⁻¹ relative to **INT4B-L4**). Therefore, we

cannot exclude either of them. The calculational results have been added in the revised Supporting Information and the relevant references have been cited in the revised manuscript.

Based on DFT conclusions and the reported literature, we can conclude that the differences in mechanism can mainly be attributed to the electronic effects of the ligands. When the ligand donates more electrons to the metal center, the charge on the metal center increases, leading to a decrease in the bond energy of the Ni-C bond, making it easier to break. However, electron-deficient ligands weaken the charge on the metal center, facilitating their η^2 -coordination with alkenes, thus enabling isomerization reactions.

2. *There is one instance of “cis/trans tautomerization”, which needs correcting to “cis/trans isomerization”.*

Answer: This error has been corrected.

3. *Regarding the large tables of computed energies and coordinates in the Supporting Information, the suggestion was not to actually remove these from the SI, but rather, to encourage the authors to make the raw data available as an attachment (either as Supporting Information or in a data repository) that can be downloaded by readers, and then examined using computational chemistry software. Apologies if this was not 100% clear.*

Answer: Thank you for your suggestion. We still place the calculated energies and coordinates in the Supporting Information so that they can be easily accessed by interested readers.

4. *The authors have satisfactorily answered most of the questions from the reviewers regarding substrate scope (what works and what doesn't work), and about what happens when certain suggested experiments are carried out. However, not all of this information is provided in the paper (or SI) itself. I am not suggesting that the authors provide full explanations in the paper, i.e., reproducing exactly what is in the rebuttal letter, but for certain cases, I think it would be useful to add a brief comment in the paper to explain. E.g. for points 1 and 2 of Reviewer 2, and point 1 of Reviewer 3.*

Answer: Thank you for your suggestion. We have added corresponding explanations in the main text as required.

Response to point 1 of Reviewer 2: the following sentence was added in the manuscript.

“Notably, the direct reductive coupling of aryl bromides and trifluoromethyl alkenes in this catalytic system is well suppressed.”

Response to point 2 of Reviewer 2: the following sentence was added in the manuscript.

*“Despite extensive attempts, cis-arylation cyclization/cross-coupling products have not yet been obtained using **L9** as a ligand.”*

Response to point 1 of Reviewer 3: the following sentence was added in the manuscript.

“Nonetheless, alkyl-substituted alkynes and terminal alkynes failed to produce the expected products.”

Reviewer #2 (Remarks to the Author):

This revised manuscript has addressed all issues raised by all of the reviewers, and thus is recommended to be accepted for publication without further revisions.

Reviewer #3 (Remarks to the Author):

Kong and co-workers have tried best to fulfill the referees' suggestions on this manuscript. Thus, I suggest its acceptance for publication in nature communications

Reviewer #4 (Remarks to the Author):

This paper is a revised manuscript previously submitted to Nature Communications as a full paper. The authors have added necessary computational trials to address all my concerns in my initial comments, although there's a trivial mistake in their rebuttal ("Figure 8B" on page 14 should be Scheme 8B). **Now the computational results are more detailed and reliable, and the writing looks more readable. I think the present manuscript is suitable for publication on Nature Communications as it stands.**